# Protein Oxidation Biomarkers and Myeloperoxidase Activation in Cerebrospinal Fluid in Childhood Bacterial Meningitis

**DOI:** 10.3390/antiox8100441

**Published:** 2019-10-01

**Authors:** Emilie Rugemalira, Irmeli Roine, Julia Kuligowski, Ángel Sánchez-Illana, José David Piñeiro-Ramos, Sture Andersson, Heikki Peltola, Manuel Leite Cruzeiro, Tuula Pelkonen, Máximo Vento

**Affiliations:** 1Children’s Hospital, Helsinki University Hospital, Stenbäckinkatu 9, 00029 Helsinki, Finland; sture.andersson@helsinki.fi (S.A.); heiheikkipeltola@gmail.com (H.P.); tuula.pelkonen@hotmail.com (T.P.); 2Faculty of Medicine, University of Helsinki, Yliopistonkatu 4, 00014 Helsinki, Finland; 3Faculty of Medicine, University Diego Portales, Manuel Rodrigues Sur 333, 8370109 Santiago Region Metropolitana, Chile; irmeli.roine@gmail.com; 4Health Research Institute La Fe, Avenida Fernando Abril Martorell 106, 46026 Valencia, Spain; julia.kuligowski@uv.es (J.K.); asanchezillana@gmail.com (Á.S.-I.); josedavidpineiro@gmail.com (J.D.P.-R.); maximo.vento@uv.es (M.V.); 5Hospital Pediátrico David Bernardino, Rua Amilcar Cabral, Luanda, Angola; mcruzeiro1@gmail.com; 6Division of Neonatology, University and Polytechnic Hospital La Fe, Avenida Fernando Abril Martorell 106, 46026 Valencia, Spain

**Keywords:** oxidative stress, protein damage, myeloperoxidase, bacterial meningitis, developing countries

## Abstract

The immunological response in bacterial meningitis (BM) causes the formation of reactive oxygen and nitrogen species (ROS, RNS) and activates myeloperoxidase (MPO), an inflammatory enzyme. Thus, structural oxidative and nitrosative damage to proteins and DNA occurs. We aimed to asses these events in the cerebrospinal fluid (CSF) of pediatric BM patients. Phenylalanine (Phe), para-tyrosine (p-Tyr), nucleoside 2′-deoxiguanosine (2dG), and biomarkers of ROS/RNS-induced protein and DNA oxidation: ortho-tyrosine (o-Tyr), 3-chlorotyrosine (3Cl-Tyr), 3-nitrotyrosine (3NO₂-Tyr) and 8-oxo-2′-deoxyguanosine (8OHdG), concentrations were measured by liquid chromatography coupled to tandem mass spectrometry in the initial CSF of 79 children with BM and 10 without BM. All biomarkers, normalized with their corresponding precursors, showed higher median concentrations (*p* < 0.0001) in BM compared with controls, except 8OHdG/2dG. The ratios o-Tyr/Phe, 3Cl-Tyr/p-Tyr and 3NO₂-Tyr/p-Tyr were 570, 20 and 4.5 times as high, respectively. A significantly higher 3Cl-Tyr/p-Tyr ratio was found in BM caused by *Streptococcus pneumoniae*, than by *Haemophilus influenzae* type b, or *Neisseria meningitidis* (*p* = 0.002 for both). In conclusion, biomarkers indicating oxidative damage to proteins distinguished BM patients from non-BM, most clearly the o-Tyr/Phe ratio. The high 3Cl-Tyr/p-Tyr ratio in pneumococcal meningitis suggests robust inflammation because 3Cl-Tyr is a marker of MPO activation and, indirectly, of inflammation.

## 1. Introduction

Despite improved vaccination programs and antibiotics, bacterial meningitis (BM) remains the tenth leading cause of global under-5 deaths [1]. If not fatal, it can lead to serious sequelae such as cognitive deficit, hearing loss, motor deficit, seizures, visual impairment, or hydrocephalus [2]. The neuronal damage is caused by the direct effect of the microorganisms entering the subarachnoid space (SAS) and the host’s strong inflammatory reaction [3,4]. When recognizing bacterial components in the SAS, Toll-like and other receptors of the innate immune system cause activation of nuclear factor kappa B (NF-κB). This in turn regulates the expression of pro-inflammatory cytokines and chemokines [5].

Consequently, polymorphonuclear leukocytes (PMNs) are attracted to the focus of infection. Stimulated PMNs activate their nicotinamide adenine dinucleotide phosphate (NADPH) oxidase complex (NOX2), which, using oxygen as a substrate, produces high amounts of superoxide anion (O_2_•−). Superoxide anion dismutates into hydrogen peroxide (H_2_O_2_), which by the action of myeloperoxidase (MPO) further combines with protons and chloride ions, leading to the formation of hypochlorous acid (HClO, Figure 1). Hypochlorous acid plays an important role in killing bacteria and, during inflammation, in modifying biomolecules in the host [6,7]. Moreover, in the presence of transition metals, hydrogen peroxide can be converted into highly reactive ferryl species (Fenton reaction) [8].

In addition to NOX2, another antimicrobial system of the PMN is the inducible nitric oxide synthase pathway (iNOS) that produces nitric oxide (NO). Peroxynitrite (ONOO^–^), also a pro-inflammatory and cytotoxic reactive oxidant, is generated by the reaction of nitric oxide and superoxide anion [5].

Oxidative/nitrosative stress is characterized by the generation of increased amounts of reactive oxygen/nitrogen species (ROS/RNS) that exceed the capacity of the cellular antioxidant defense system. Under these circumstances, the pro-oxidant status leads to structural and/or functional oxidative/nitrosative damage to proteins, DNA and lipids that can be assessed by the detection of specific metabolites in different biofluids [9,10,11]. Hence, phenylalanine (Phe) is used as a sentinel for protein damage because, in contrast to other amino acids, it has only one physiological end-product, para-tyrosine (p-Tyr, Figure 2).

In situations of oxidative stress (Figure 2), ferryl species and peroxidases oxidize Phe to ortho-tyrosine (o-Tyr) and meta-tyrosine (m-Tyr), whereas peroxynitrite or hypochlorous acid attack p-Tyr producing 3-nitrotyrosine (3NO₂-Tyr) or 3-chlorotyrosine (3Cl-Tyr), respectively. O-Tyr, 3NO₂-Tyr and 3Cl-Tyr are considered reliable biomarkers of oxidative protein damage [12,13]. In addition, the 3Cl-Tyr/p-Tyr ratio is a reliable biomarker of tissue inflammation. 8-Oxo-2′-deoxyguanosine (8OHdG) is produced by the oxidation of the nucleoside 2′-deoxyguanosine (2dG), and the ratio 8OHdG/2dG is employed as an index of oxidative DNA damage [11].

In BM, the host’s inflammatory response is potentially a life-threatening phenomenon where oxidative/nitrosative stress plays a major pathophysiological role [5,14]. For better understanding of the underlying pathophysiological mechanisms we aimed to asses which metabolites derived from oxidative/nitrosative stress and MPO activation best characterize the oxidative/nitrosative damage and inflammatory response in the cerebrospinal fluid (CSF) in children with BM.

## 2. Materials and Methods

### 2.1. Study Design and Patients

The patient data was collected from a prospective single-center study carried out in the Pediatric Hospital of Luanda, Angola, from 2005 to 2008 [15]. Children aged 2 months to 13 years with suspected BM (*n* = 723) were included. BM was considered confirmed if the child with signs and symptoms of BM had positive CSF culture, positive CSF polymerase chain reaction (PCR), positive blood culture, or at least two of the following criteria: CSF pleocytosis > 100 cells/mm^3^ (predominantly polymorphs), a positive Gram-stain result, positive latex-agglutination test, or serum C-reactive protein (CRP) > 40 mg/L. All children were treated with cefotaxime for seven days but were randomized to receive it either by slow continuous infusion or by boluses every six hours for the first 24 h. In addition, the patients received high-dose paracetamol or placebo for the first 48 h. Oral glycerol was given to all children as adjuvant treatment. The details of the original study are explained elsewhere [14].

The current post-hoc analysis focuses on a sub-cohort including 79 patients with confirmed BM from whom a CSF sample taken at presentation to the Pediatric Hospital of Luanda was available. The samples were stored at –80 °C until processing. Ten control samples from 2018, which were anonymous pool samples from children of whom a central nervous system infection was suspected but eventually excluded, were provided by Helsinki University Hospital, Department of Virology.

### 2.2. Standards and Reagents

The analytical standards employed for the determinations were purchased from Sigma-Aldrich (St. Louis, MO, USA) for o-Tyr, Phe, 3NO_2_-Tyr, 3Cl-Tyr, p-Tyr, 8OHdG and 2dG (>96% *w/w* purity). The isotopically labeled compounds used as internal standards were deuterated phenylalanine (Phe-D_5_) from CDN Isotopes (Pointe-Claire, QC, Canada); 8-Oxo-2′-deoxyguanosine-^13^C^15^N_2_ (8OHdG-^13^C^15^N_2_) and 2′-deoxyguanosine-^13^C^15^N_2_ (2dG-^13^C^15^N_2_) from Toronto Research Chemicals (Toronto, ON, Canada); and deuterated para-tyrosine (p-Tyr-D_2_) from Cambridge Isotope Laboratories (Tewksbury, MA, USA). The purity of all isotopic labeled compounds was > 98% *w/w*. Milli-Q^®^ grade water (>18.2 MΩ) was used from a Milli-Q^®^ Integral system (Darmstadt, Germany). Solvents used (Methanol (CH_3_OH) and acetonitrile (CH_3_CN)) (LC-MS grade) were obtained from JT Baker (Deventer, The Netherlands) and sodium hydroxide, phosphoric acid, and formic acid (HCOOH) were purchased from Panreac Química (Barcelona, Spain).

### 2.3. Sample Preparation and Analysis Employing Liquid Chromatography Coupled to Tandem Mass Spectrometry (LC-MS/MS)

The CSF samples were thawed on ice, homogenized for 30 s employing a Vortex^®^ mixer, followed by centrifugation (15 min, 4 °C, 10,000 g). Fifty microliters of the CSF supernatant was diluted with 50 µL of CH_3_OH:H_2_O:HCOOH (15:85:0.1 *v/v*).

A working solution containing Phe, p-Tyr, o-Tyr, 3Cl-Tyr, 3NO_2_-Tyr, 2dG, and 8OHdG was prepared by mixing the corresponding volumes of individual stock solutions prepared in H_2_O (0.1% *v/v* HCOOH). Standard solutions used for calibration, covering the concentration ranges shown in Table 1, were prepared from the working solution by serial dilution. A mixture of internal standard solutions (IS) containing Phe-D_5_, p-Tyr-D_2_, 8OHdG-^13^C^15^N_2_ and 2dG-^13^C^15^N_2_ was prepared by mixing the corresponding volumes of working solutions in H_2_O (0.1% *v/v* HCOOH).

All biomarkers were quantified simultaneously, employing an LC-MS/MS system according to previously validated methods [12,16] with slight modifications. The Acquity-Xevo TQ-S system (Waters, Milford, MA, USA) was employed for the analysis, operating in positive electrospray mode (ESI^+^). The ESI interface parameters were: 3.50 kV, source temperature: 120 °C, desolvation temperature: 300 °C, N_2_ flux for cone and desolvation: 25 and 680 L h^−1^, respectively.

Multiple reaction monitoring (MRM) was employed as a method of tandem mass spectrometry with all dwell times of 5 ms, ensuring a minimum of 10 data points per peak. The MRM instrumental parameters are summarized in Table 1. Chromatographic separation was carried out on an Acquity UPLC BEH C8 (2.1 × 100 mm, 1.7 μm) reversed phase column from Waters under a CH_3_CN (0.1% *v/v* HCOOH):H_2_O (0.1% *v/v* HCOOH) binary gradient. The gradient ran as follows: from 0.0 to 1.25 min 1% *v/v* CH_3_CN (0.05% *v/v* HCOOH) (i.e., channel B) and from 1.25 to 3.0 min %B increased to 98%. The return to initial conditions was achieved at 3.75 min and conditions were maintained for 0.75 min for system re-equilibration. Flow rate, column temperature, and injection volume were 400 µL min^–1^, 55 °C and 5 µL, respectively. During LC-MS/MS analysis, samples were stored at 4 °C in the autosampler. MassLynx Mass Spectrometry Software (version 4.1) from Waters (Waters, Milford, MA, USA) was employed for data acquisition and processing. The standards for calibration were analyzed in the same batch as the samples. Linear response curves for each analyte were calculated, employing internal standards as indicated in Table 1. Signals obtained from samples were interpolated in the corresponding calibration lines for obtaining absolute concentration values.

### 2.4. Statistical Analysis

Phe, p-Tyr, 2dG and their derivatives o-Tyr, 3Cl-Tyr, 3NO₂-Tyr, and 8OHdG were measured from the CSF samples taken on admission. The protein and DNA biomarker concentrations were normalized with their corresponding precursors. The values below the limit of quantification (LOQ) were replaced by their corresponding 0.5 × LOQ.

All data were analyzed with Statview^®^ software, version 5.0.1 (SAS institute, Cary, NC, USA). The Mann-Whitney U test and Kruskal-Wallis test were used when appropriate.

### 2.5. Ethics

After approval of the study protocol by the Luanda Children’s Hospital ethics committee in 2005 the study was registered as ISRCTN62824827. Each child’s legal guardian signed an informed consent prior to the start. The study was conducted according to the principles of the Declaration of Helsinki.

## 3. Results

Of the 79 BM patients (38 female, median age 12 months), the most common identified causative agent was *Streptococcus pneumoniae* (*n* = 40), followed by *Haemophilus influenzae* type b (Hib, *n* = 24) and *Neisseria meningitidis* (*n* = 11). The other agents (*n* = 4) were *Escherichia coli*, *Proteus vulgaris*, *Pseudomonas* and *Salmonella*. Twenty-eight (35%) children died and the case fatality rate for pneumococcal, meningococcal and Hib meningitis was 40% (16/40), 18% (2/11) and 29% (7/24), respectively. The patient characteristics are listed in Table 2.

The concentrations of Phe (Rho 0.353, *p* = 0.0018), p-Tyr (Rho 0.449, *p* < 0.0001), o-Tyr (Rho 0.432, *p* = 0.0002), 3Cl-Tyr (Rho 0.359, *p* = 0.0005), 3NO₂-Tyr (Rho 0.399, *p* = 0.0004) and the ratio o-Tyr/Phe (Rho 0.403, *p* = 0.0004) correlated positively with the CSF leukocyte count in BM patients. A negative correlation existed between the concentrations of Phe (Rho −0.26, *p* = 0.02449, p-Tyr (Rho −0.413, *p* = 0.0003), 3Cl-Tyr (Rho −0.261, *p* = 0.022) and CSF glucose.

The median CSF levels of protein and DNA oxidation biomarkers and the ratios to their precursors in BM patients and controls are listed in Table 3. The BM patients showed a significantly higher median concentration (*p* < 0.0001) in all other biomarkers except the 8OHdG/2dG ratio when compared to the control patients. The ratios o-Tyr/Phe, 3Cl-Tyr/p-Tyr and 3NO₂-Tyr/p-Tyr were 570, 20 and 4.5 times as high, respectively (Table 3).

When comparing the three most common types of BM, pneumococcal meningitis differed significantly with a higher 3Cl-Tyr/p-Tyr ratio from Hib (*p* = 0.031) or meningococcal disease (*p* = 0.002) (Table 4).

## 4. Discussion

Protein oxidation biomarkers and especially the o-Tyr/Phe ratio in CSF samples of children with BM reflected the presence of oxidative/nitrosative stress as compared to children without a central nervous system infection. Furthermore, both 3Cl-Tyr, a stable compound that indicates inflammation, and the 3Cl-Tyr/p-Tyr ratio may thus serve as a fingerprint for the MPO-catalyzed oxidation reaction (Figure 1 and Figure 2).

According to different animal models [4,5,14,17] and a few human studies [10,18,19] ROS/RNS are formed in activated PMNs during the inflammatory response of the host once bacteria have reached the SAS. This cascade probably plays an important role in the pathophysiology of BM and thus, it is essential to clarify the characteristics and extent of oxidative stress in human BM. Interestingly, in experimental animal models of pneumococcal meningitis, radical scavengers and antioxidants have alleviated the intracranial complications and neuronal injury [4,5,14].

Direct measurement of ROS/RNS, antioxidant activities and different oxidative damage biomarker concentrations have been used to measure the degree of oxidative/nitrosative stress in the human BM studies [10,18,19]. Firstly, the direct measurement of the highly reactive and short-lived radicals is demanding, and secondly the measurements of antioxidation activities are not only dependent of oxidative stress but of other regulatory mechanisms [20]. Thus, our approach was to measure stable oxidation byproducts.

The high CSF leukocyte count, diagnostic for BM, correlated with the concentrations of Phe and its derivatives. Thus, to eliminate the influence of the expected lower CSF leukocyte count of our control patients, we normalized the biomarker concentrations to their precursors instead of employing absolute concentrations when comparing BM to non-BM. We interpreted the increased Phe and p-Tyr in BM patients as linked to the elevated CSF protein concentration, which unfortunately, due to the limited resources in Luanda, we were unable to determine. Another perspective to consider is the alteration of amino acids, and the fluctuation in their levels in situations of inflammation. Elevated Phe concentrations and Phe/Tyr ratios has been reported in patients with chronic conditions, such as HIV-1 infection or cancer, with a background of immune activation and inflammation [21].

Pneumococcal meningitis is one of the most severe forms of BM, where the case fatality ranges from 16% to 37%, and neurological sequelae develop in 30% to 52% [22]. The outcome of BM probably relates to the severity of inflammation, especially in the SAS [22]. Here, pneumococcal meningitis, when checked at presentation to hospital, showed the highest 3Cl-Tyr/p-Tyr ratio in the CSF and the difference was significant when compared with Hib or meningococcal meningitis. This finding, indicative of robust inflammation, accords with the known disease severity of this type of BM [23]. The low CSF glucose concentration in BM and its correlation with 3Cl-Tyr and its precursors further support this finding—low CSF glucose is a well know prognostic indication of dismal outcome [23].

The number of studies investigating MPO activity in childhood BM is limited. Miric et al. found that the MPO activity measured in CSF was enhanced on admission, 3 to 5 days later, and before discontinuation of BM treatment, compared to a control group with other types of meningeal irritation [24]. An experimental study in neonate rats with induced *Streptococcus agalactiae* meningitis [25] showed a remarkable MPO activity from 24 to 96 h in the hippocampus and from 6 to 96 h in the cortex. Our series indirectly confirmed MPO activation, and to our knowledge, is the first where 3Cl-Tyr/p-Tyr was determined as a biomarker for protein damage and as an indirect indicator for MPO activation in CSF in childhood BM.

3NO_2_-Tyr is used as a biomarker for peroxynitrite formation. An increase in the concentration of 3NO₂-Tyr in the initial CSF samples was found previously in adult BM [26]. High CSF concentrations of 3NO₂-Tyr were also associated with an unfavorable outcome of BM. However, 3NO_2_-Tyr can be a product of different metabolic pathways and thus does not serve as a specific biomarker for ONOO^−^ formation [12].

In one study, the CSF 8OHdG concentration was significantly higher in early BM than in aseptic meningitis or in children without any meningitis [27]. Furthermore, the 8OHdG concentration decreased along with the clinical and laboratory improvement of BM [27]. We assessed the 8OHdG/2dG ratio and did not observe an increment compared with the control patients. The lack of comparison to the precursor, in addition to the differing method applied (e.g., ELISA) in the biomarker quantification might explain this discrepancy between studies. Nonetheless, the high specificity of the LC-MS/MS method enables multianalyte quantification of the biomarkers in a single sample and is thus highly valuable with limited volume samples [28].

We acknowledge limitations in our study. This was a post-hoc analysis in which the data were collected from a study originally designed for other purposes. For this reason, a sufficiently large CSF sample was available from only around 11% of the entire series. Samples from BM patients had been kept ultra-frozen for several years, while the control samples were more recent. However, all samples were initially processed and stored at –80 °C until analysis without additional freeze-thaw cycles to minimize bias due to sample degradation. In addition, the patient and control groups were of different origin, which raises the possibility of differing genetics influencing the results.

## 5. Conclusions

We conclude that protein oxidation biomarkers are elevated in childhood bacterial meningitis as a sign of oxidative/nitrosative stress observed most clearly in the o-Tyr/Phe ratio. A recent review on oxidative stress, aging and disease stated that studies based on observations made with bacteria, plants and mammals suggest that abnormal tyrosine isomers (o-Tyr, m-Tyr) are not just markers of oxidative stress but also mediators of its effects [29]. In addition, a high observed 3Cl-Tyr/p-Tyr ratio indicates MPO activation and could serve as a biomarker for grading the severity of inflammation and, orientate the clinician towards a pneumococcal infection.

Additional studies are needed for a better understanding of the host’s complex inflammatory response in BM, and in determining the oxidative/nitrosative stress in relation to a dismal outcome. Up to now, no clinical trials have been performed in BM patients testing adjuvant drugs interfering with the intracranial complications and neuronal damage caused by ROS/RNS.

## Figures and Tables

**Figure 1 antioxidants-08-00441-f001:**
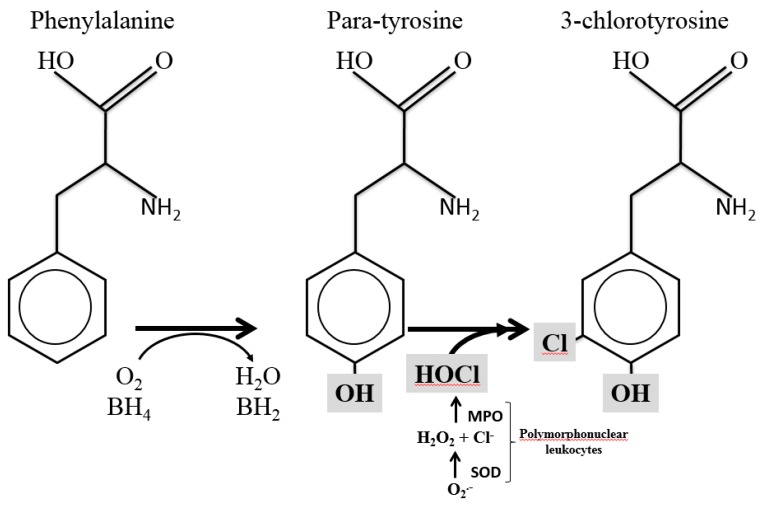
Synthesis of 3-chlorotyrosine. MPO = myeloperoxidase; H_2_O_2_ = hydrogen peroxide; O_2_•− = superoxide; Cl^−^ = chloride; SOD = Superoxide dismutase.

**Figure 2 antioxidants-08-00441-f002:**
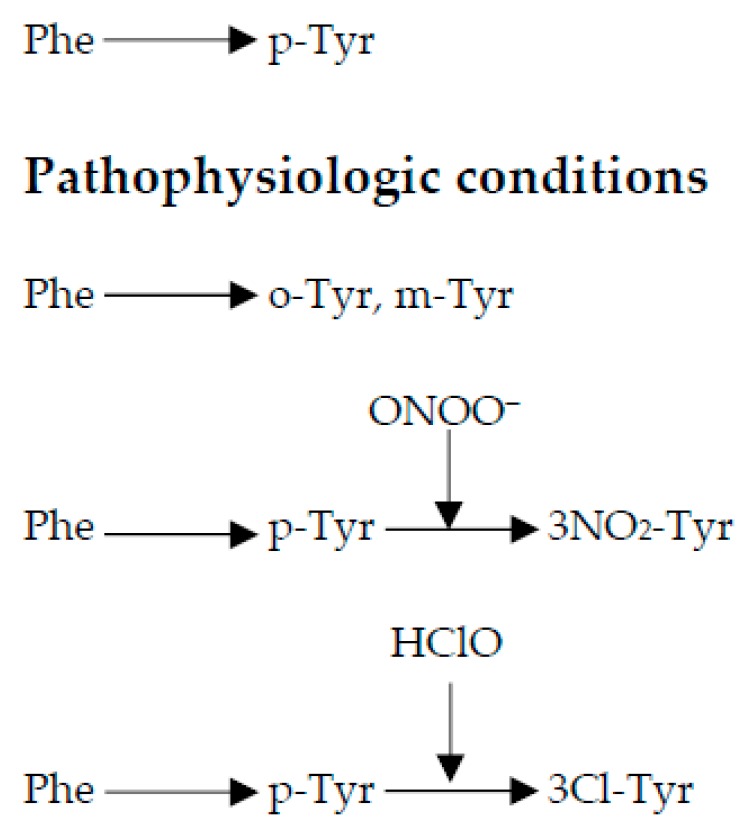
Under physiological conditions, phenylalanine (Phe) is enzymatically oxidized to para-tyrosine (p-Tyr) by the action of phenylalanine hydroxylase. Under pathophysiological conditions, Phe is oxidized by ferryl species and peroxidases into ortho-tyrosine (o-tyr) and meta-tyrosine (m-tyr), and p-Tyr by peroxynitrite (ONOO^−^) into 3-nitrotyrosine (3NO_2_-Tyr) or by hypochlorous acid (HClO) into 3-chlorotyrosine (3Cl-Tyr).

**Table 1 antioxidants-08-00441-t001:** Analytical parameters and figures of merit.

Analyte	Retention Time ± s [min]	Calibration Range	*R* ^2^	*y* = a + b*x*		*m/z* Parent Ion	Cone [V]	Daughter Ion		Internal Standard
				a [nm^–1^]	b [nM]			CE [eV]	*m/z* Quantification	
Phenylalanine (Phe)	2.32 ± 0.01	0.2–400 µM	0.954	16.4	0.01	166.1	20	20	91.0	Phe-D_5_
Para-tyrosine (p-Tyr)	1.01 ± 0.01	0.2–400 µM	0.999	38.8	0.03	182.1	20	10	91.0	p-Tyr-d_2_
Ortho-tyrosine (o-Tyr)	1.80 ± 0.01	1–2000 nM	0.999	–0.1	0.15					p-Tyr-d_2_
3-chlorotyrosine (3Cl-Tyr)	1.90 ± 0.01	2–4000 nM	0.999	–2.1	1.68	216.0	20	15	170.0	Phe-D_5_
3-nitrotyrosine (3NO_2_-Tyr)	2.33 ± 0.01	1–2000 nM	0.995	8.0	2.87	227.1	25	10	181.0	Phe-D_5_
2-deoxyguanosine (2dG)	1.45 ± 0.03	1–2000 nM	0.999	–0.3	0.41	268.0	25	15	152.0	2dG-^13^C^15^N
8-oxo-2′-deoxyguanosine (8OHdG)	2.04 ± 0.02	1–500 nM	0.999	0.1	0.35	284.0	30	15	168.0	8OHdG-^13^C^15^N
Phenylalanine-d_5_ (Phe-D_5_)	2.32 ± 0.01	-	-		-	171.5	30	20	125.0	-
p-Tyrosine-d_2_ (p-Tyr-D_2_)	1.01 ± 0.02	-	-		-	184.1	20	10	138.1	-
2-Deoxyguanosine-^13^C^15^N (2dG-^13^C^15^N)	1.45 ± 0.02	-	-		-	271.0	15	10	155.0	-
8-Oxo-2′-deoxyguanosine-^13^C^15^N (8OHdG-^13^C^15^N)	2.04 ± 0.03	-	-		-	287.0	30	15	171.0	-

CE: Collision Energy.

**Table 2 antioxidants-08-00441-t002:** Baseline characteristics of bacterial meningitis (BM) patients in Luanda with analyzed admission cerebrospinal fluid (CSF) samples.

VARIABLE	BM (*n* = 79)
Age in months, median (IQR)	12 (7–42)
Weight for age below—2SD	19 (24%)
Duration of illness days, median (IQR)	4 (3–7)
Previous antibiotics *	30/74 (41%)
Glasgow coma score, median (IQR)	11 (7–14) ᵃ
Another focus of infection	19 (24%)
Cerebrospinal fluid	
Leukocyte count (×10⁶/L), median (IQR)	1740 (353–3515)
Glucose concentration (mg/dL), median (IQR)	17 (9–26) ᵇ
Blood	
CRP ^#^ on day 1 or 2 (mg/L), median ** (IQR)	154 (81–161) ᶜ
Glucose (mg/dL), median (IQR) ***	85 (62–111) ᵈ
Hemoglobin day 1 or 2 (g/dL), median (IQR)	7.5 (6–9) ^e^
Causative agent	
*Streptococcus pneumoniae*	40/79 (51%)
*Haemophilus influenzae* type b	24/79 (30%)
*Neisseria meningitidis*	11/79 (14%)
Other bacteria	4/79 (5%)

ᵃ *n* = 78, ᵇ *n* = 78, ᶜ *n* = 34, ᵈ *n* = 75, ^e^
*n* = 77; * Number of patients of whom data were available are shown.; ^#^ CRP stands for C-reactive protein; ** When CRP level exceeded 160 mg/L it was marked as 161 mg/L; *** Lowest glucose on day 1.

**Table 3 antioxidants-08-00441-t003:** Comparison of the admission median cerebrospinal fluid (CSF) concentrations (IQR) in nmol/L of biomarkers of ROS/RNS-mediated stress to proteins, DNA and inflammation between children with and without bacterial meningitis. Mann-Whitney U test.

VARIABLE	BM, Luanda (*n* = 79)	Control, Helsinki (*n* = 10)	*p* Value	Ratio BM/Contol
Phenylalanine (Phe)	88,346 (51,535−166,316)	6558.0 (5249.76−8473)	< 0.0001	13.5
Para-tyrosine (p-Tyr)	64,214 (31,197−152,125)	13,239 (10,096−17,677)	< 0.0001	4.9
Ortho-tyrosine (o-Tyr)	162.12 (65.23−2194.9)	0.02 (0.020)	< 0.0001	8100
3-chlorotyrosine (3Cl-Tyr)	423.34 (134.84−1311.95)	4.155 (4.155)	< 0.0001	102
3-nitrotyrosine (3NO₂-Tyr)	90.48 (59.37−135.47)	2.745 (1.181−4.61)	< 0.0001	32.7
2′deoxiguanosine (2dG)	303.57 (91.32−1329.69)	0.768 (0.768−3.538)	< 0.0001	395
8-oxo-2′-deoxyguanosine (8OHdG)	3.895 (3.895−17.778)	0.043 (0.043)	< 0.0001	90.6
Ratio 3Cl-Tyr/p-Tyr	0.007 (0.003−0.022)	3.531 × 10^−4^ (2.498 × 10^−4^−0.001)	< 0.0001	19.8
Ratio o-Tyr/Phe	0.002 (0.001−0.013)	3.4995 × 10^−6^ (2.85 × 10^−6^−6.665 × 10^−5^)	< 0.0001	572
Ratio 3NO₂-Tyr/p-Tyr	0.001 (0.001−0.002)	2.235 × 10^−4^(1088 × 10^−4^–4.444 × 10^−4^)	< 0.0001	4.5
Ratio 8OHdG/2dG	0.025 (0.005−0.063)	0.056 (0.012−0.056)	0.428	0.45

**Table 4 antioxidants-08-00441-t004:** Comparison of the admission median cerebrospinal fluid (CSF) concentrations (IQR) in nmol/L of markers of ROS/RNS-mediated stress to proteins, DNA and inflammation in children with different etiologies of bacterial meningitis. Kruskal-Wallis test.

	*Streptococcus pneumoniae*	*Haemophilus influenzae*	*Neisseria meningitidis*	*p* Value
VARIABLE	*n* = 40	*n* = 24	*n* = 11	
Phenylalanine (Phe)	84,788 (43,702−131,709)	90,950 (58,433−172,710)	166,203 (46,372−219,475)	0.4454
Para-tyrosine (p-Tyr)	59,871 (39,143−135,736)	53,641 (28,876−112,299)	153,636 (30,053−226,175)	0.4902
Ortho-tyrosine (o-Tyr)	155.77 (60.51−1973.3)	109.175 (50.75−1212.4)	687.97 (95.335−6987.8)	0.3539
3-chlorotyrosine (3Cl-Tyr)	719.72 (289.6−2654.7)	317.17 (104.79−1251.7)	246.17 (101.36−497.2)	0.0568
3-nitrotyrosine (3NO₂-Tyr)	77.26 (52.49−125.0)	91.91 (79.130−158.63)	107.28 (57.94−144.99)	0.39
2′-deoxiguanosine (2dG)	521.94 (104.37−6199.1)	161.98 (93.29−536.1)	84 (54.26−281.38)	0.0213
8-oxo-2′-deoxyguanosine (8OHdG)	4.103 (3.895−23.83)	3.895 (3.895−8.655)	3.895 (3.895−8.619)	0.3452
Ratio o-Tyr/Phe	0.001 (0.001−0.011)	0.001 (0.001−0.013)	0.011 (0.002−0.033)	0.2968
Ratio 3Cl-Tyr/p-Tyr	0.012 (0.005−0.028)	0.004 (0.002−0.018)	0.002 (0.002−0.033)	0.0021
Ratio 3NO₂-Tyr/p-Tyr	0.001 (0.001−0.002)	0.002 (0.001−0.003)	0.001 (0.001−0.002)	0.287
Ratio 8OHdG/2dG	0.02 (0.004−0.059)	0.025 (0.011−0.058)	0.046 (0.026−0.072)	0.1647

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
