# Peer review of "Protein Oxidation Biomarkers and Myeloperoxidase Activation in Cerebrospinal Fluid in Childhood Bacterial Meningitis"

_antioxidants, 2019, doi:10.3390/antiox8100441_

Round 1

Reviewer 1 Report

Authors described protein oxidation biomarkers and myeloperoxidase activation in cerebrospinal fluid in childhood bacterial meningitis. They conclude that high 3Cl-Tyr/p-Tyr ratio in pneumococcal meningitis suggests a robust inflammation, since 3Cl-Tyr is a marker of MPO activation and indirectly of inflammation.

This is a potentially interesting paper, however, authors provide misleading and incorrect description of hydroxyl (OH) radical under these pathological conditions. The sections below must be corrected.  

Authors must be noted that OH-radical does not have specific products since it reacts immediately with diffusion controlled rate constant with any neighbouring C-H group. The formation of ortho-tyrosine and meta-tyrosine must be explained by either direct oxidation by peroxidase or reaction with a Ferryl species such as >Fe=O. 

Authors must delete and modify following sections:

Page 2: "Moreover, in the presence of transition metals, hydrogen peroxide can be converted into highly reactive and bactericidal hydroxyl radical (˙OH)."

Delete bactericidal hydroxyl radical (˙OH). It is wrong! OH radical does not have any specific function since it is absolutely chemically or biologically NON-specific.

Use "can be converted into highly reactive ferryl species or oxidized heme. 

Page 3 and Figure 2 must be changed.

Page 3: "In situations of oxidative stress (Figure 2), hydroxyl radical oxidizes Phe to ortho-tyrosine (o-Tyr) and meta-tyrosine (m-Tyr)..."

Remove "hydroxyl radical oxidizes" and add "ferryl species and peroxidases oxidize..."

Figure 2: "Under pathophysiological conditions, Phe is spontaneously oxidized by hydroxyl radical (·OH) into ortho-tyrosine (o-tyr) and meta-tyrosine (m-tyr)"

Remove from Figure 2 "spontaneously oxidized by hydroxyl radical (·OH)" and replace with "oxidized by peroxidases and ferryl species"

Add reference showing correct role of tyrosine oxidation by heme/peroxidases and/or ferryl species such as 

Proc Natl Acad Sci U S A. 2004 Mar 23;101(12):4003-8.

Antioxid Redox Signal. 2013 Jun 10;18(17):2264-73. Toxicol In Vitro. 2014 Aug;28(5):847-55.  

Author Response

Point 1: Authors must be noted that OH-radical does not have specific products since it reacts immediately with diffusion controlled rate constant with any neighbouring C-H group. The formation of ortho-tyrosine and meta-tyrosine must be explained by either direct oxidation by peroxidase or reaction with  Ferryl species such as >Fe=O.

Response 1: Hydroxyl radical is highly reactive against DNA, proteins, and lipids, but is not bactericidal as stated by the reviewer. Its half-life is so short that it only causes alterations of very nearby molecules. In the presence of iron or other transition metals the Fenton reaction (1) ensues and may enhance its damaging effects .

The suggested revisions have been made to the paper

(1.) Radi R. Nitric oxide, oxidants, and protein tyrosine nitration. Proc Natl Acad Sci U S A. 2004; 101:4003-4008.

Reviewer 2 Report

Overall an interesting study. Well written. Only comment is that the  Legend of figure 1 seems misplaced or not complete. 

Author Response

Point 1: Legend of figure 1 seems misplaced or not complete. 

The abbreviations to Figure 1 seem to have shifted to page 3. 

Reviewer 3 Report

The article "Protein Oxidation Biomarkers and Myeloperoxidase Activation in Cerebrospinal Fluid in Childhood Bacterial Meningitis" examines oxidative stress markers in children with bacterial menigitis. Here the authors have shown that oxidative stress markers are elevated in the CSF of these patients and that there are some differences seen between different pathogens. The paper is well written and I have no issue recommending this for publication.

Line 256 would read better if plant and mammalian was replaced with plants and mammals.

Author Response

Point 1. Line 256 would read better if plant and mammalian was replaced with plants and mammals.

Response 1. Corrected to plants and mammals.